# A Systematic Review on the Improvement of Cd Stress Tolerance in Ramie Crop, Limitations and Future Prospective

Adnan Rasheed [1], Hongdong Jie [1], Pengliang He [1], Xueying Lv [1], Basharat Ali [2], Yushen Ma [1], Hucheng Xing [1], Saad Almari [3], Rehab O. Elnour [4], Muhammad Umair Hassan [5], Syed Faheem Anjum Gillani [6] and Yucheng Jie [1,*]

1. College of Agronomy, Hunan Agricultural University, Changsha 410128, China; adnanbreeder@yahoo.com (A.R.); jhd20210218@stu.hunau.edu.cn (H.J.); hpl888@stu.hunau.edu.cn (P.H.); sx20210341@stu.hunau.edu.cn (X.L.); mys9204@stu.hunau.edu.cn (Y.M.); xhcsoldier@163.com (H.X.)
2. Department of Agricultural Engineering, Khwaja Fareed University of Engineering and Information Technology, Rahim Yar Khan 62400, Pakistan; dr.basharat@kfueit.edu.pk
3. Department of Biology, College of Science, King Khalid University, Abha 61413, Saudi Arabia; saralomari@kku.edu.sa
4. Biology Department, Faculty of Sciences and Arts, King Khalid University, Abha 61413, Saudi Arabia; relgezouly@kku.edu.sa
5. Research Center on Ecological Sciences, Jiangxi Agricultural University, Nanchang 330045, China; muhassanuaf@gmail.com
6. Gansu Provincial Key Lab of Arid Land Crop Science, Lanzhou 730070, China; faheemgillani33@gmail.com
* Correspondence: ibfcjyc@vip.sina.com

**Abstract:** Cadmium (Cd) is a non-essential, highly phytotoxic metal and damages ramie plant growth and development even at low concentrations. Ramie is one of the most significant crops in China, with excellent fiber quality and immense industrial importance. Planting Cd-tolerant ramie cultivars can prevent yield loss on contaminated soil. Previously, significant efforts have been made to develop Cd tolerance in ramie. However, the Cd tolerance mechanism is still not fully understood; hence, breeding industrial crops is critical to tackling the ongoing challenges. Cd tolerance is a complex genetic mechanism requiring high-level molecular studies to clarify the genes network. Genetic studies have identified several Cd-tolerant genes in ramie, which led to the development of several ramie cultivars suitable to grow on toxic soils; however, due to the continuous rise in Cd toxicity, potent molecular tools are critical in modern-day breeding programs. Genetic engineering, and transcriptome analysis have been used to develop abiotic stress tolerance in ramie, but QTL mapping and clustered regularly interspaced short palindromic repeats (CRISPR) are rarely studied. However, studies are still limited in addressing this issue. This review critically elaborated on using QTL mapping, transcriptomes, transcription factors, CRISPR/Cas9, and genetic engineering to enhance Cd tolerance in ramie. These genes/QTL should be transferred or edited into sensitive cultivars using genetic engineering or CRISPR/Cas9. CRISPR/Cas9 is highly recommended because it provides targeted gene editing in ramie, its use is limited and can address the research gaps, and it would revolutionize the field of agriculture. Limitations, gaps, and future potential are briefly discussed. This review paper presents new clues to help future researchers comprehensively understand Cd tolerance in ramie and develop tolerant cultivars for industrial purposes.

**Keywords:** ramie; Cd stress; tolerance; genes; marker-assisted selection; CRISPR/Cas9

## 1. Introduction

The significant rise in levels of heavy metal ions such as Cd, arsenic (As), and lead (Pb) in soils poses a constant threat to human and plant life [1–4]. These metals are naturally present in our environment in low concentrations [5,6]. Human-caused activities are a significant cause of the rising concentration of heavy metals in soil [7–10]. Cd stress is the most devastating abiotic stress that has led to the huge loss of crop growth and yields

in large areas. Cd enters the food chain and causes an imbalance in food security [11,12]. Cd toxicity is a significant threat to crops. It is toxic even at low concentrations [13] and has no biological role in crop growth and development [14]. Anthropogenic activities include the atmospheric deposition of combustion emissions, mining, sewage sludge, and Cd-containing fertilizers [15]. Cd is ranked seventh in the top twenty toxic heavy metals list and is classified as a group 1 carcinogen [16]. $Cd^{2+}$ is a divalent form of Cd in soils at concentrations typically ranging from 0.1–1.0 mg $kg^{-1}$ [17]. Cd stress interferes with crops' morphological, physiological, and biochemical traits. Cd stress mainly inhibits seed germination, reducing total plant weight, the number of leaves, and root length, ultimately leading to total plant death [18]. It encourages osmotic stress in plants by reducing leaf-relative water content, transpiration, and stomatal conductance [19,20]. Cd stress reduces the uptake of essential nutrients and leads to inappropriate plant growth [21,22]. Ramie is called China grass, and it is considered the most critical fiber crop next to cotton and is characterized by its rapid growth, shoot biomass, and excellent root system [23]. Ramie can be harvested three times a year. Ramie is mainly cultivated in China, India, and Pacific Rim countries. Ramie cultivation dates back to 5000 years ago in Southern China, with a high fiber yield next to cotton [23]. Toxic levels of Cd have led to a reduction in root length, shoot length, fiber yield, and enzymatic activities [12,24]. Meanwhile, it also reduces the chlorophyll contents, plant water relation, nutrients uptake, biomass, quality, protein contents, and damage to DNA [25]. Cd pollution is more terrible in the Hunan Province of China than in most parts of China [26]. China has small arable land per capita, but the food demand is too high and requires the effective and safe utilization of heavy metals-contaminated arable land to sustain food security. In recent years, numerous studies have focused on safe utilization, including soil improvement, selection, breeding varieties with low capacity for the absorption of heavy metals, and agronomic management [27].

Previous studies have revealed that ramie crop has robust tolerance to Cd stress and can accumulate Cd ions from soil; hence, the use of ramie to prevent the Cd contamination of agricultural soils is of practical significance [27–29]. More significantly, ramie can accumulate a comparatively large number of heavy metals in its above-ground portion [30]. Ramie is a relatively tolerant crop against heavy metals stress, especially Cd stress, as it absorbs Cd from heavy metals-contaminated soils [27,30,31]. An earlier study showed that ramie has a certain degree of tolerance to soil Cd stress at a concentration of ≤20 mg/kg [32]. Ma et al. [33] treated the two ramie cultivars (Dazhuhuangbaima and Zhongzhu 1) with different Cd levels of 0, 25, and 75 mg $kg^{-1}$ for 30 days and concluded that Zhongzhu 1 had a higher degree of tolerance to Cd stress, as indicated by Cd enrichment in all organs and Cd content was mainly observed in the cell wall of both varieties. In this regard, breeders have employed conventional breeding techniques to enhance the Cd tolerance in ramie [34]. Still, the complex genetic mechanism of Cd tolerance hindered the expansion of these techniques. On the other hand, hormonal applications significantly mediated ramie growth under Cd stress [35]. Previous research studies have shown that ramie has a potential tolerance against Cd stress, which can be exploited to develop Cd-tolerant cultivars; however, there is still limited information about the complete genetic mechanism of ramie.

Considering these issues, it is essential to tackle the ongoing issue of Cd stress by using potential molecular tools. In this regard, the QTL mapping analysis method is one of the most promising ways of improving tolerance in ramie [36]. Still, this technique has limited applications in ramie, and this gap should be covered in future studies. On the other hand, TFs analysis revealed several genes involved in Cd tolerance [30,37] and can be used to develop tolerant cultivars using genetic engineering or editing using the CRISPR/Cas9 tool. All TFs belong to different families, which have still not been identified because of the diverse nature of the gene network. Several gene families must be explored for their possible role in Cd tolerance in ramie [37,38]. CRISPR/Cas9 is a novel gene editing tool that offers targeted editing of the desired gene in crops [39]. Still, this tool is not used for Cd tolerance in ramie, which presents an excellent opportunity to target Cd-sensitive genes

to generate Cd-tolerant mutants in future studies. As one of the most significant fiber crops, ramie growth and yield are continuously threatened by rising toxic levels of Cd stress. Until now, many studies have been conducted to unpin the molecular mechanism of Cd tolerance in ramie; however, a detailed review on this aspect has not been published to motivate breeders to carry out additional studies. This is the first comprehensive review paper on Cd tolerance in ramie, which will serve as a valuable source of information to execute future research studies. The bioremediation of Cd-polluted sites is considered effective and reliable because ramie is a super accumulator of Cd.

## 2. Effects of Cd Stress and Genetic Mechanisms

In plants, Cd toxicity causes leaf chlorosis, reduced growth rates, respiration and photosynthesis, enhanced oxidative damage, and decreased nutrient uptake capability [40,41]. Ramie's growth and development are badly affected by Cd stress. In previous studies, different ramie genotypes were evaluated to assess the effects of Cd stress. Cd stress affected the plant height, shoot dry weight, and root dry weight [35,42]. Ramie seedlings were subjected to Cd stress (3 and 7 mg$1^{-1}$) for 10 days to investigate its effects on lipid membrane and chlorophyll synthesis. Cd stress-induced lipid peroxidation, as indicated by an enhanced malondialdehyde content (MDA) in leaves and roots. The prolonged exposure of ramie to Cd stress decreased antioxidant enzyme activity [24]. Zhu et al. [43] studied the effects of Cd stress on the growth parameters of different ramie cultivars. They concluded that Cd stress negatively affects the above-ground raw tissues and fiber yield [43]. Another ramie cultivar, Chuanzhu, was treated with Cd stress at a concentration of 50 mg/L, and it was noticed that plant height (PH), root length (RL), and biomass were reduced [44]. Higher Cd concentration reduces photosynthesis (Figure 1) and affects the reproductive stage in ramie [45]. Cd stress caused sugar degradation, the inactivation of antioxidant enzyme activity, protein oxidation, and growth inhibition in ramie [46]. Increasing Cd concentration affected the stem, leaves, and internal cell organs in ramie [47]. As mentioned earlier, Cd is highly toxic to ramie even at low concentrations. Some of the micro-regional studies showed that the ramie cultivar is significantly affected by Cd stress at 25 mg kg$^{-1}$ concentration. Cd retained in stems and leaves greatly reduced the plant biomass [48]. Although several studies have indicated the toxic effects of Cd stress on ramie crops, there is still a need to expose the plant to extreme Cd stress to investigate the effects on biochemical and molecular functions. These studies would help us to adopt a better protective mechanism to counter the adverse effects of Cd stress through the screening and training of genotypes [49].

In the last decade, numerous studies have focused on the Cd tolerance mechanisms at the physiological, biochemical, and molecular levels [50,51]. Transcriptome and proteomics approaches have been adopted to understand the molecular factors involved in Cd tolerance [52,53]. Tolerance to Cd stress in ramie is a complex genetic mechanism controlled by many genes [49]. The Cd tolerance in ramie varies among the genotypes and germplasm. Ying et al. [54] evaluated the 269 ramie germplasm resources under the Cd stress condition. The results indicated a significant difference among the ramie germplasm regarding Cd accumulation and transport. The data showed the different genetic factors involved in ramie tolerance to Cd stress, which can be analyzed using molecular breeding tools rather than conventional tools [54]. Plants also have varied and complex defensive systems for resisting Cd stress [55].

Moreover, antioxidant enzymes, ascorbate peroxidase (APX), superoxide dismutase (SOD), and catalase (CAT) are critical for scavenging ROS [56–58]. The complex genetic mechanism of Cd tolerance cannot be unfolded using conventional breeding methods because of their time-consuming and costly nature [59–61]. Despite all these efforts, the Cd-tolerant nature of ramie has not been thoroughly studied. The effects of Cd stress on fiber quality and genetic material should be further investigated to avoid potential loss. Hence, it would be better to make a complete genome sequence of ramie germplasm to characterize all possible genes and proteins involved in Cd tolerance in ramie and use them

in Cd stress breeding programs. In addition to this, germplasm with Cd-tolerant ability can be fully exploited for Cd-tolerant genotype development. The activation of systematic resistance plays a crucial role in countering Cd stress in ramie and can be enhanced using Cd stress conditions. Investigations of the morphological, physiological, and biochemical mechanisms of Cd tolerance are not fully understood and need further studies to identify the potential traits involved in Cd tolerance in ramie.

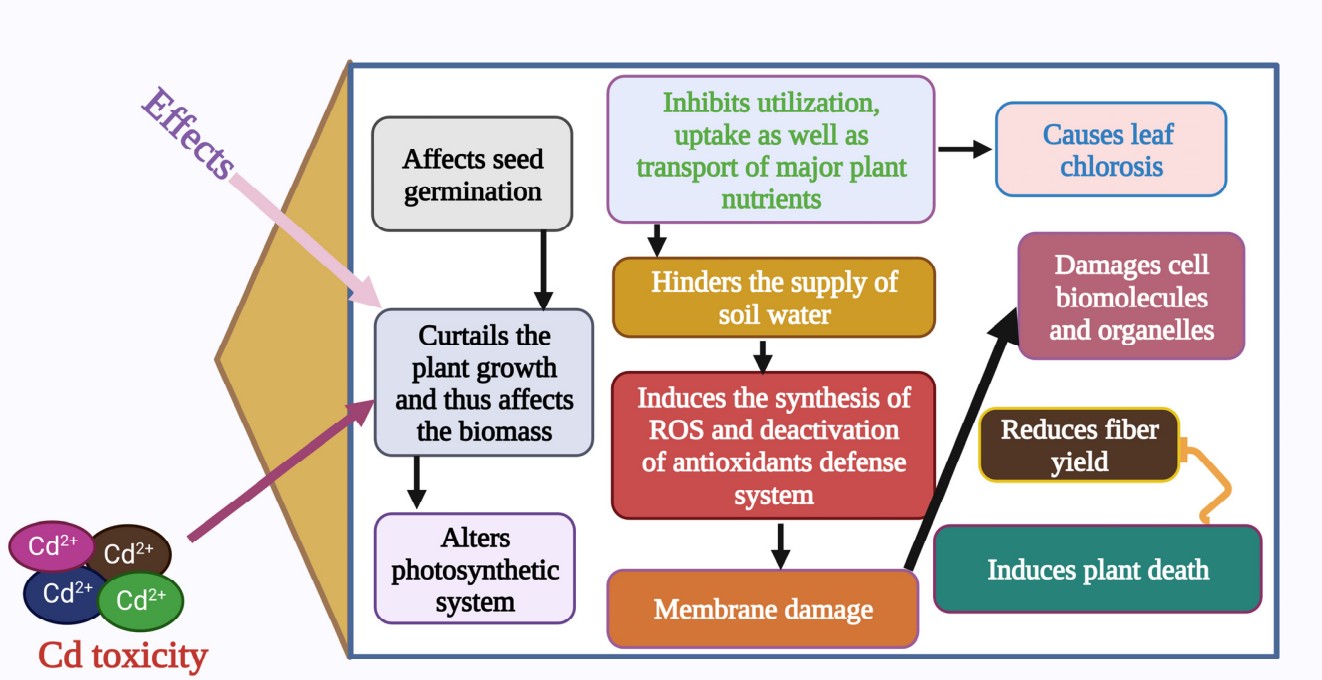

**Figure 1.** Effects of Cd toxicity on ramie growth and development. Cd toxicity leads to a reduction in seed germination, seedling growth, biomass, and photosynthesis. Cd toxicity reduces the uptake of nutrients and water. In the same way, Cd toxicity induces the production of reactive oxygen species (ROS) and reduces the activity of antioxidant enzymes. Cd toxicity causes damage to membranes and biomolecules. Cd toxicity reduces yield and causes plant death. This Figure is created with Biorender.com.

## 3. Agronomic Approaches to Enhance Cd Tolerance in Ramie

Plant hormones or growth regulators play a crucial role in defense response against heavy metals (HMs) stress, especially Cd stress [62,63] (Figure 2). Each hormone's mechanism of action may differ for each crop [64]. It has been determined that multiple hormones may regulate a single process, and simultaneously, different processes are controlled by a single hormone [65]. Plant hormones such as auxin, abscisic acid (ABA), cytokinin (CK), ethylene (ET), and jasmonic acid (JA) are essential plant hormones that are vital for plant growth and development, and are also involved in crosstalk with other hormones [66–68]. Hormones in very low concentrations control cell membrane permeability, secondary metabolites, and growth, as well as the reproduction of plants [69,70]. Cd stress retards plant growth and accumulates in below- and above-ground parts of the plant. Cd stress causes a delay in seed germination, and seeds may vary in their response [71]. Plants react to heavy Cd stress depending on the concentration of Cd stress. The role of these hormones has been investigated in many crops, including ramie, which has the ability of Cd phytoextraction [72].

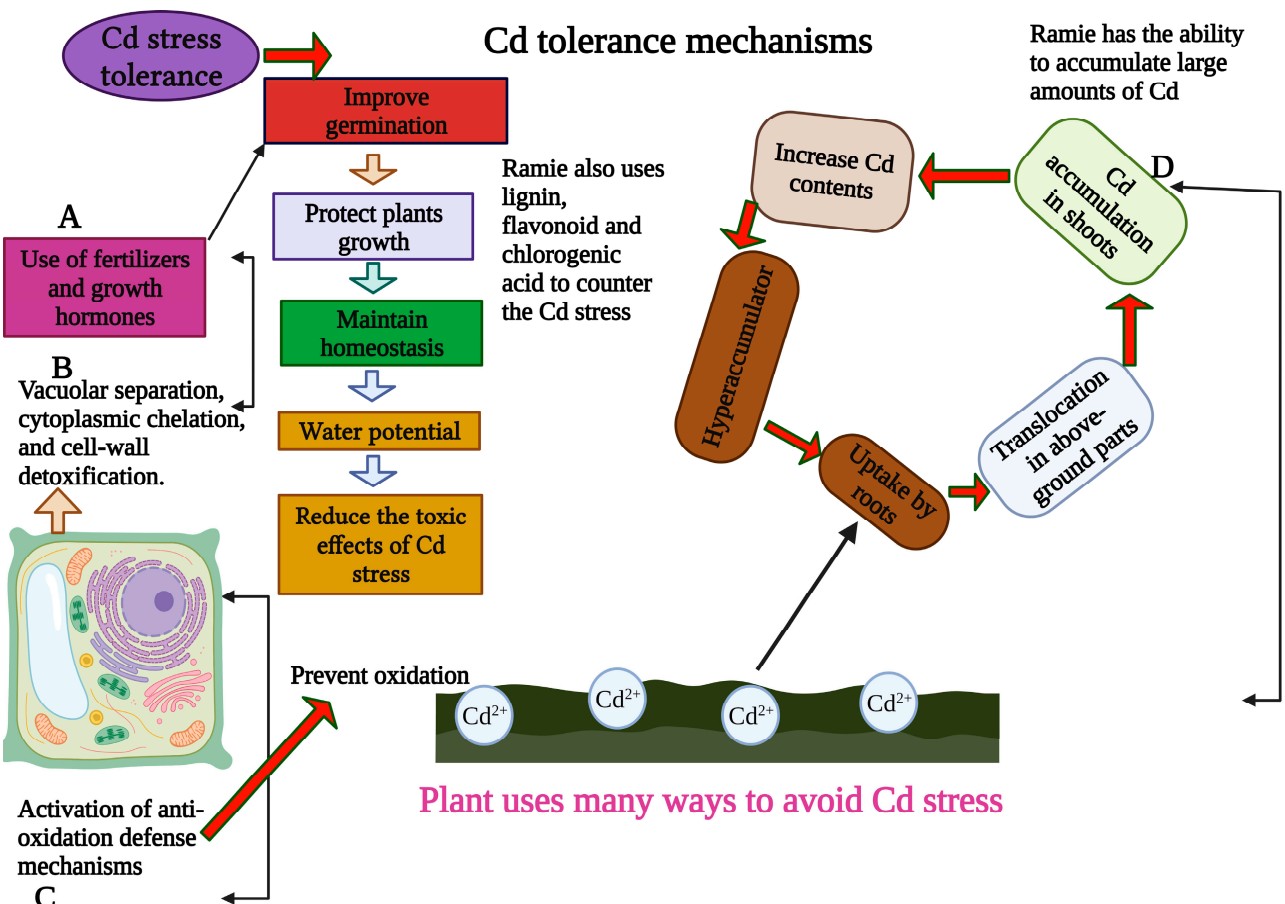

**Figure 2.** The ways to prevent Cd stress in ramie. The plant mainly prevents Cd stress by detoxification and chelation. The ramie plant also uses antioxidant defense mechanisms to avoid Cd stress. This Figure is created with Biorender.com.

ABA is a powerful hormone controlling plant tolerance to Cd stress in ramie. Chen et al. [73] conducted a study to investigate the effect of ABA on ramie under Cd stress. Directly adding ABA to the cultural solution promoted ramie growth and development under Cd stress, increased Cd enrichment, and indicated its regulatory role in enhancement or stress tolerance and as a potential agronomic agent [73]. Similarly, another research team led by He et al. [74] studied the effect of different growth regulators on the Cd content in ramie. The results showed that a foliar spray of hormones reduced the Cd content in underground parts of ramie, which showed the potential role of these growth regulators in increasing Cd tolerance [74].

Auxin is well known for its signaling function and plays a crucial role in life developments such as cell division, embryogenesis, meristem, and organ development [75,76]. Auxin plays a significant role in stress response, organ formation, and vascular development [77]. A recent study showed that Cd at a concentration of 500 mM had variable effects on hormonal function. The content of JA reduces, and ABA increases after exposure to Cd stress. Meanwhile, indole acetic acid (IAA) content decreases after increased Cd content in *Koelreuteria paniculata* [78]. Similarly, ABA content increases in soybean after exposure to Cd stress for 24 h and decreases after 140 h [79]. Chaca et al. [79] revealed that plants mainly produce JA and ABA hormones under Cd stress. Usually, phytohormones enhance plant tolerance to Cd due to their participation as signaling molecules and encourage plant growth by expressing different metabolites, genes, and pigments [80,81]. Earlier work has demonstrated that plant hormones regulate Cd tolerance in many crops, including rice [82], wheat [83], and *Arabidopsis* [84]. These crops respond to Cd stress by hormonal action. The priming of barley seed with GB3 reduces the toxic effects of Cd stress on

coleoptile growth [85]. Gibberellin improves ramie growth, development, and metabolic processes [86,87]. The study indicated that brassinosteroids (BR) and gibberellin (GA) have different effects on the enrichment and transport of Cd in ramie. The results indicated that GA-1 enhanced the Cd content of the aboveground ramie to three times more than that of the control and decreased the Cd content of the underground ramie by 54.76%. This research study offers an effective technique to advance the capability of ramie to adsorb heavy metals [74].

Furthermore, the pretreatment of seeds of maize with SA and IAA enhances the growth of coleoptiles under Cd stress [88]. In another study, SA at a concentration of 500 mM increased the antioxidant defense system [89] and mitigation of the influences of Cd stress on plant growth and development in maize [90]. It has also been shown that SA use reduces the Cd stress in roots [91]. In barley plants, SA curbs the consequences of Cd stress on organic reserves, inhibits the impairment of plant growth due to Cd, and triggers Cd tolerance mechanisms [92]. Kalai et al. [92] described that SA at concentration of 600 mM enhanced growth and development in barley seedlings and decreased Cd stress. These reports showed very little information about using hormones to mitigate Cd stress in crops. It is strongly recommended to conduct more studies to investigate the regulatory role of plant growth regulators in alleviating Cd toxicity in ramie to develop a suitable approach for Cd-contaminated soils, which are the primary habitat of ramie.

Spermidine is a polyamine compound situated in ribosomes and living tissues and has numerous metabolic roles within plants [93–95]. Spermidine has been a significant polyamine in mitigating Cd stress in many crops [96]. Ramie was used to investigate the Cd tolerance mechanism with various treatment durations and the effect of the up-regulation of antioxidant capability by spermidine treatment. The outcomes exhibited that short-term (0–7 days) Cd stress caused the enrichment of pigment content, soluble sugars, and the activation of antioxidants, temporarily reducing the level of MDA, the primary oxidative stress marker. Spd treatment noticeably improved the soluble sugar content and condensed glutathione (GSH) in ramie leaves after short-duration Cd stress, while it showed no beneficial effects on other antioxidants. Long-term (0–15 days) Cd stress may lead to growth retardation related to Cd increment, protein sugar degradation, a rise in MDA, and the inactivation of antioxidants. Spd treatment reduced long-term Cd toxicity by reducing Cd content, stabilizing cellular macromolecules such as protein and sugar, and preventing the inactivation of catalase (CAT) and peroxidase (POD). These findings showed that Spd uses could be dynamic in the Cd tolerance's rise by regulating different traits [46].

Another study used the combined exogenous application of Spd and calcium (Ca) to study the effects on Cd stress in ramie. The results exhibited that using 5 mM Ca meaningfully reduced Cd toxicity in ramie by reducing Cd content, discouraging $H_2O_2$ and MDA content, enhancing plants' dry weight and chlorophyll contents, and changing the actions of total superoxide dismutase (SOD). Additionally, ramie accumulated Cd and was affected by more toxic effects of Cd stress by applying 1 mM Ca or Cd. Severe Cd toxicity could be reduced by adding exogenous Spd via a surge of plant growth and the decrease of the Cd-induced oxidative stress. Generally, the effects of 1 mM Ca and Spd seemed greater than other treatments in the ramie plants under Cd stress, with a more excellent Cd accumulation capability. They assessed Cd stress tolerance in ramie [97]. Sun et al. [34] studied the effects and mechanisms of IAA and Spd on Cd accumulation in ramie to broaden the understanding of Cd tolerance. The results showed that the application of Spd and IAA increased the activity of antioxidant enzymes (Table 1). IAA and Spd application could meaningfully reduce the oxidative stress encouraged by Cd in ramie crop. Hence, the application of Spd, IAA, and GA can be combined to increase the Cd tolerance in ramie crops [34].

**Table 1.** Role of different hormones, polyamines, nutrients, and metalloids to enhance Cd tolerance in ramie.

| Cd Stress | Hormones, Polyamines, Metalloids, and Nutrients | Effects | References |
|---|---|---|---|
| 30 mg kg$^{-1}$ | Spd and IAA 100 μm | Increased the activity of antioxidant enzymes | [34] |
| 6 and 9 mg/L | Se, 1.2 μmol/L | Improve the action of POD and SOD and control methylation of DNA in ramie leaves | [25] |
| 1 mM | Ca 5-mM, and Spd 1 mM | Reduced MDA contents, enhancing plants' dry weight, and chlorophyll contents | [97] |
| Cd 30 μM | Spd 0.1 mM | Decreased Cd content, stabilized cellular macromolecules like protein and sugar | [46] |

Selenium (Se) is one of the most potent metalloids widely used to enhance crop tolerance to Cd stress. Se has also been used in ramie to mitigate the toxic effects of Cd stress. Ramie plants were grown in hydroponic conditions and were separately or instantaneously treated with Se spray, 1.2 μmol/L, and Cd stress 6 and 9 mg/L. At low Se levels, the activity of SOD was improved by 35.34% under 6 or 9 mg/L Cd stress, POD was improved by 12.45%, and the level of DNA methylation was reduced by 10.70%, correspondingly. The results established that low Se-level spray on ramie leaves could improve the action of POD and SOD, and control DNA methylation in ramie leaves. Hence, future studies must explore the effects of this hormone on functions of other antioxidants such as CAT and APX to improve the Cd tolerance in ramie [25]. In an earlier study, ethylenediaminedisuccinic acid (EDDS) and ethylenediaminetetraacetic acid (EDTA) were added to potted soil containing ramie plants. The chelants were found to decrease Cd content in roots by 15% and 12% in EDTA- and EDDS-treated plants. Hence, chelate-induced phytoremediation is an effective technique for the remediation of Cd-contaminated soils [98]. Similarly, in another study, the Cd accumulation and translocation mechanism in ramie was studied by adding ethylenediaminetetraacetic acid (EDTA) or nitrilotriacetic acid (NTA). The results demonstrated that EDTA and NTA could increase Cd Phyto-availability in soil, transport Cd from the roots to the aboveground tissues, and increase leaf Cd accumulation. This study highlights the control of Cd accumulation in ramie [99]. The above results indicate that more studies should be conducted to investigate the role of GA, CK, and ET in regulating Cd tolerance by enhancing the growth of ramie plants. These hormones can enhance the antioxidant defense system and prevent ramie growth under severe Cd stress conditions. Moreover, the hormonal application should be studied at different growth stages of ramie under Cd stress conditions.

## 4. Genetic Diversity for Cd Tolerance in Ramie

The breeding of genotypes tolerant to adverse growing environments is critical for ramie existence [100]. Hence, genetic diversity has been the foundation of plant breeding since the initial days of agriculture [101,102]. It helps plant breeders to develop new genotypes that can address grower requirements, acclimatize to climate variation, and meet the growing global food demand [103]. Researchers rely on dissimilarity in crop genetics, breeding methods, and approaches to integrate genetic assortment into cultivated genotypes [104]. Crop breeders exploit genetic diversity to breed new cultivars with improved traits such as resistance to biotic and abiotic stresses and to advance the nutritional value of foods for the world population [105,106]. Plant breeders have achieved the vital task of the planned incorporation of a new genetic assortment while protecting critical economic characteristics of crops [103]. Genetic diversity can be explained as a range of genetic traits in crop species. Genetic diversity can be measured by investigating changes in the DNA nucleotide sequences in a progeny of individuals [107–109]. Genetic diversity in crops gives rise to a population's phenotypic differences and observable traits [110]. Crop genomes are naturally different, which causes more genetic and epigenetic variations in plants that

help as the sources of a large volume of genetic and phenotypic diversity, even among the cultivars of the same species [111,112]. More considerable genetic diversity in crops gives plants an outstanding capability to acclimatize to sudden abiotic stresses [113,114].

The achievement of the crop improvement target depends on capably recognizing and integrating genetic diversity from numerous plant genetic sources comprising currently grown varieties [115], newly established varieties, wild and landraces of crops, and the collection of germplasm with elite plants [116]. Numerous genomic tools and breeding approaches have enhanced the competence and accuracy of integrating genetic diversity into cultivated crop varieties. However, plant breeding still remains a time- and resource-intensive procedure [117,118]. Ramie, a significant fiber crop [119], has genetic diversity, crucial in developing tolerant cultivars [120]. Genetic diversity is indispensable for breeding programs. Ramie has excellent potential for tolerance to Cd toxicity [54]. To assess genetic diversity for Cd tolerance, 269 ramie germplasm accessions were studied under Cd-contaminated soils and significant genetic diversity was observed among the studied germplasm [54]. Ramie has been best known for its phytoremediation capability for decades. The phytoremediation capability of ramie shows that it has considerable genetic diversity for Cd-tolerant genes, which can be exploited in breeding programs. Zhu et al. [121] showed that ramie could tolerate higher Cd concentrations (100 mg/kg), prevent significant economic loss, and safeguard crop security. Results showed that an extended period would be required to improve the soils affected by Cd toxicity. Other assortments and cultivations of Cd hyperaccumulator ramie are indispensable [121]. The first report on constitutional heavy metals tolerance was presented by Yang et al. [23]. They evaluated ramie in contaminated fields with various metals, including Cd, and revealed that ramie germplasm and genotypes performed better under contaminated conditions. These results determined that ramie holds a certain degree of constitutional Cd tolerance. This germplasm can be useful for increasing diversity and transforming tolerant genes to develop Cd-tolerant ramie cultivars [23].

The results presented significant variations in Cd tolerance; these germplasm resources can be exploited for other breeding programs [23]. Later, another hydroponic and field experiment was conducted to compare ramie's superior genotypes and Cd tolerance indices. In the hydroponic culture experiment, considerable variances, and noteworthy genotypic differences in plant height (PH), leaf number per plant (LNP), shoot dry weight (SDW), and root dry weight (RDW) were observed under Cd treatments. The SDW was well associated with PH and RDW. The SDW was well associated with PH and bark thickness. This proposes that these features could be used as Cd tolerance indices in future studies [122]. She et al. [123] showed that Cd-tolerant ramie will be a great source of phytoremediation in Cd-contaminated soils. Cd contents in ramie were two to ten times higher compared to other plant species, as shown by the data collected from mining areas where the soil was heavily polluted with Cd and other metals. These ramie genotypes provide a more significant source of genetic diversity for Cd tolerance [123]. Zhu et al. [43] observed significant differences among the ramie varieties for Cd tolerance and accumulation. Results indicated that raw fiber yield and biological tissues of variety (Zhongzhu1) were higher than the other six varieties [43]. Several traits of ramie have been tested for Cd tolerance and accumulation. The significant difference in yield among the ramie cultivars showed more considerable genetic diversity and potential candidate genes for genetic engineering. The results obtained from a study indicated that the cultivar Zhongzhu No. 3 showed a high yield in Cd-contaminated soils. All varieties showed more than a 90% Cd removal ratio during the uncovering process of fibers [124]. The varieties such as Zhongzhu1 and Zhongzhu No. 3 should be used to develop new high-tolerant varieties for the phytoremediation of Cd-contaminated soils. The screening of all available germplasms of ramie to identify the potential Cd tolerant plant material for further breeding programs is critical.

Therefore, it is proposed that the screening of Cd-tolerant ramie genotypes would lead to the identification of tolerant genes for Cd-tolerant breeding programs. The screening ramie genotypes can be enhanced via breeding tools such as introduction, mass selection,

and hybridization (Figure 3) to enhance Cd tolerance in the future when the crop faces more deadly threats of Cd toxicity. Molecular breeding tools are, however, more potent, and reliable in protecting the diversity and integrating the Cd-tolerant genes into elite cultivars. We recommend collecting the ramie germplasm from different locations and screening it for its tolerance against Cd stress.

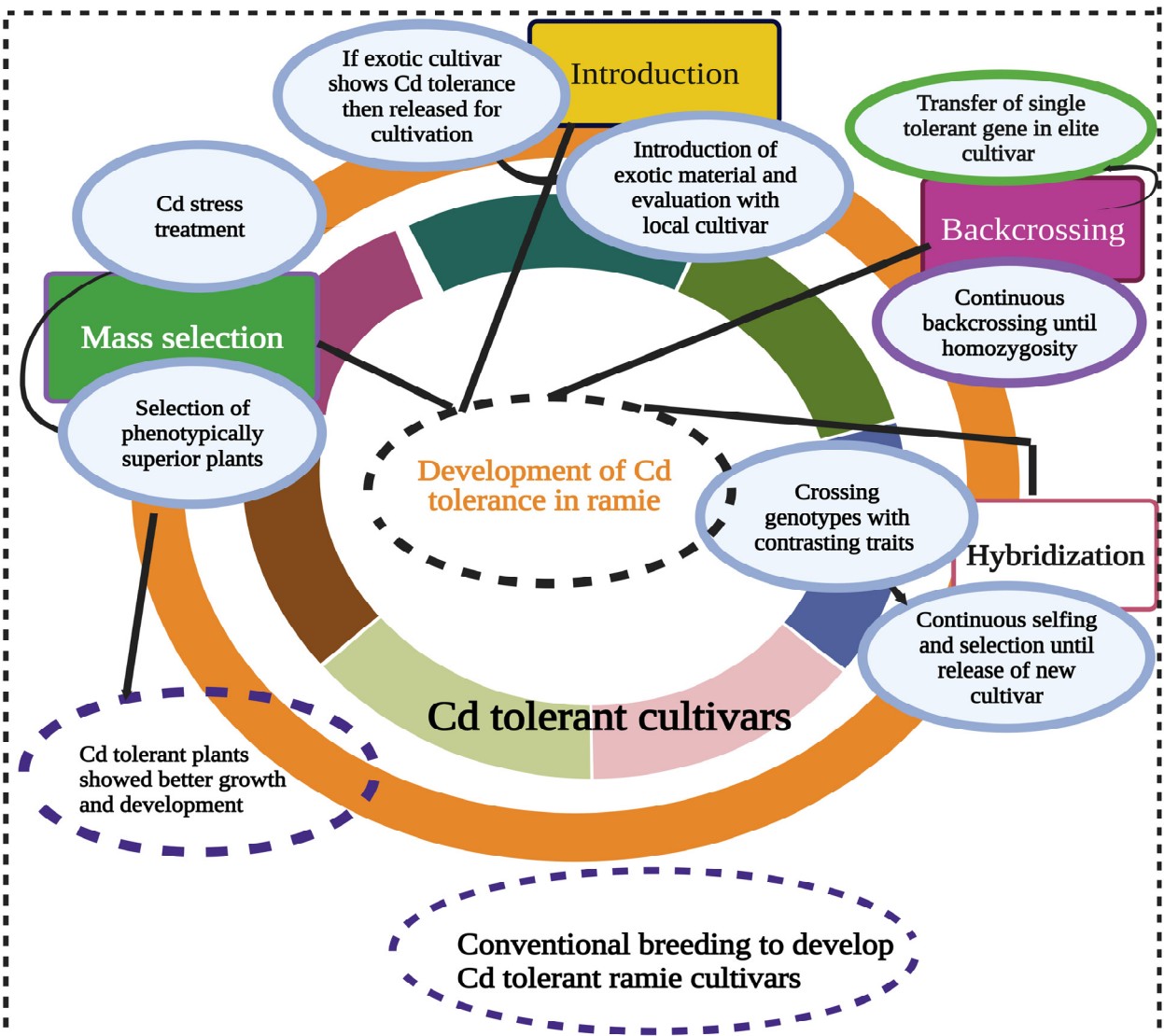

**Figure 3.** Developing Cd-tolerant ramie cultivar using conventional breeding methods such as introduction, mass selection, hybridization, and backcrossing. These breeding methods have great potential to develop Cd tolerance suitable for growth in Cd-contaminated soils. This Figure is created with Biorender.com.

## 5. QTL Mapping for Cd Tolerance in Ramie

The highly toxic, metal-like Cd and its uptake by crops are significant constraints for modern-day agriculture [125]. To dissect the genetic mechanism and regulation of Cd tolerance in crops, QTL mapping is one of the most productive approaches. QTL mapping is frequently used to recognize candidate genes related to phenotypic characters [125]. Current development in genetic and genomic resources, including genome sequencing, diversity analysis, transcriptome characterization, the construction of genetic maps, and the development of transgenic cultivars, provide new prospects to improve the genetic makeup of the ramie cultivars for better output and higher resistance to biotic and abiotic stresses. These parameters have been significantly reviewed in ramie [31]. Unfortunately, the

information on QTL mapping in ramie is negligible, which needs urgent studies to screen the population for Cd tolerance and identify the QTL to be used in the QTL pyramiding program [36,126]. Researchers have identified QTL in ramie, which were mainly related to fiber yield-related traits, lignin biosynthesis, and flowering time [126–129]. Choosing an ideal mapping population is critical in identifying dominant regions controlling Cd tolerance [125]. It is believed that the evaluation of ramie populations under different levels of Cd stress will lead to the identification of several Cd-tolerant QTL, which can be integrated into elite cultivars using QTL pyramiding. QTL mapping studies must be expanded to develop the Cd-tolerant ramie cultivars. The above results show that there are minimal studies on QTL mapping for Cd tolerance in ramie. Hence, it is essential to develop the mapping populations of ramie to detect the genomic regions regulating Cd tolerance to accelerate plant breeding.

## 6. Genetically Engineered Ramie for Cd Tolerance

Developing tolerance to Cd stress is one of the main objectives of crop breeding to sustain the crops growth and development on Cd-contaminated soils (Figure 4) [130,131]. Genetic studies have been conducted to develop tolerant cultivars in many crops using the genes for specific stress-related pathways [132,133]. Despite the significant progress in this field, the use of this technique in ramie is still minimal because of the availability of little knowledge regarding the complete genetic mechanism of Cd tolerance [134]. These hindrances limit the ramie's capability to perform better on contaminated soils [134]. Using genetic engineering, ramie's capabilities to mitigate the toxic effects of Cd stress may have considerably increased [134]. Genetic engineering can potentially transfer the tolerant gene to develop Cd-tolerant plants [130,135]. The growth of genetically engineered ramie is a suitable approach to sustain growth in Cd-affected soil.

Although gene expression and regulation caused by Cd stress are not fully understood, She et al. [134] established the ramie root gene expression database grown as either in control or under Cd stress (100 uM). A total of 3887 genes were detected with different expression patterns. The expression pattern shows that 2883 were up-regulated and 1004 genes were downregulated. The biological function of these genes was determined by gene ontology analysis. In total, 15 genes with differential expression patterns were selected, and 12 showed consistent expression patterns. These genes could be used to develop new transgenic ramie plants via genetic engineering [134]. These genes can be potential candidates for the genetic engineering of Cd tolerance in ramie [134]. GE is a dominant method that may accelerate the development of new plant lines with traits mandatory in phytoremediation [136]. Transgenic plants with various genes from diverse sources have been developed to increase heavy metal tolerance, enabling durable heavy metal tolerance [137,138].

Genetic engineering has been a potent tool for gene modification for decades [139,140]. This technique has led to several Cd-tolerant crops showing tolerance to heavy metals as well as higher yield and quality [51,141]. *Arabidopsis* is a model plant that has shown greater tolerance to Cd stress. The development of genetically engineered *Arabidopsis* provides clues about using GE in ramie to enhance Cd tolerance in future research studies [134,141]. Applications of genetic engineering have been witnessed in many crops such as rice [51], wheat [142], *Arabidopsis* [141], and sorghum [143] (Table 2), which showed a considerable enhancement in Cd tolerance and gave solid clues for the use of this technique in ramie crops. Recent studies have shown that transcription factors (TFs), transcriptome, and many other genes are potential targets for genetic engineering to enhance Cd tolerance in crops [143,144]. The success of GE depends on the selection of *Agrobacterium* vectors, which play a central role in the transformation and expression of the transgene in our desired crop [145]. The exposure of crops to various levels of Cd stress leads to the expression of different TFs, and genes which serve as an ideal target for GE [146]. Due to the development of specific advanced gene editing methods such as CRISPR/Cas9 [147], the use of GE is decreasing and has several deficiencies [148,149], which must be addressed to solve the

Cd stress issues threatening the growth and production of economically important crops. Applications of genetic engineering are very limited in ramie. We strongly emphasize screening the ramie germplasm under Cd stress environments and identifying the potential genes to be transformed using genetic engineering. These research gaps must be filled to develop transgenic ramie cultivars that grow under Cd-contaminated soils. The wild relatives, landraces, and untapped germplasm will serve as a potential source of genetic diversity and identify novel Cd-tolerant genes in the future. However, the future of crop genome editing is bright and can meet challenges regarding food security issues and environmental hazards.

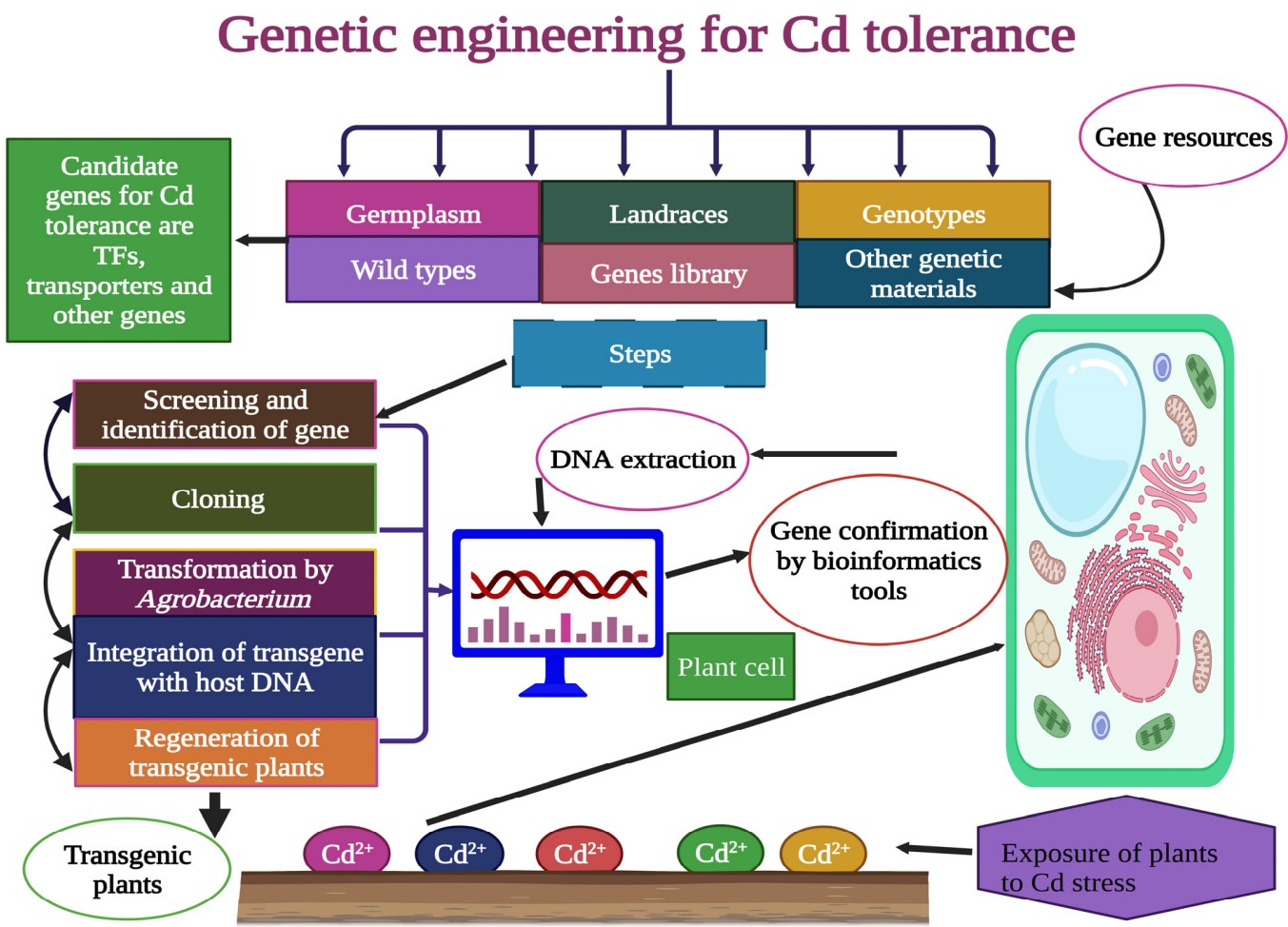

**Figure 4.** An overview of genetic engineering applications for Cd tolerance in ramie cultivars. Transgenic plants with transgene overexpression show better growth and development under Cd stress conditions. This Figure is created with Biorender.com.

**Table 2.** The exploitation of genetic engineering for Cd tolerance in significant crops.

| Crops/Plants | Gene | Role | References |
|---|---|---|---|
| Wheat | *TaSFT2L* | Enhanced root and shoot growth, chlorophyll contents, and root and shoot dry weight | [142] |
| Rice | *OsMT1e* | Enhanced shoot and dry root weight and chlorophyll contents | [51] |
| *Arabidopsis* | *ApNRAMP4* | Reduced Cd contents | [150] |
| Tobacco | *IbAKR* | Increased ability for scavenging MDA, and enhanced proline contents | [151] |
| *Arabidopsis* | *ZmOXS2b* and *ZmO2L1* | Enhanced Cd tolerance by activating the targeted gene | [141] |
| Brassica | *YCF1* | Increased plant fresh weight and biomass | [152] |

### 7. Transcription Factors (TFs) Analysis for Cd Tolerance in Ramie

Cd contamination is a significant issue in China and threatens agricultural stability [153,154]. Ramie, a fiber crop, has often been projected for use as a phytoremediation of polluted soils. Still, extreme Cd stress can critically impede ramie stem growth, decreasing its economic value [59]. To eliminate this threat, breeders must use molecular factors involved in Cd tolerance in crops and use them in molecular breeding programs [155]. TFs are essential elements that regulate gene expression and play a key role in heavy metals tolerance in plants. Plenty of TFs have been identified in response to heavy metals stress in crops; many TFs showed improved resistance, especially against Cd. The bZIP TFs family plays a critical role in heavy metals tolerance by controlling physiological processes and adaptive responses. Huang et al. [38] conducted TFs analysis and cloned a novel bZIP gene, indicated as *BnbZIP3*, from ramie plants. *BnbZIP3* transcripts were found in different tissues in ramie. *BnbZIP3* expression was induced by drought and ABA. Expression analysis showed that *BnbZIP3* might positively regulate the Cd tolerance in ramie and identified it as a potential candidate gene for future breeding programs. This study suggested that this TF, *BnbZIP3*, can be exploited in Cd-tolerant breeding in ramie in future research studies to enhance heavy metal tolerance [38]. In the same way, another significant bZIP TF, *BnbZIP2*, was identified by TFs analysis and then cloned from ramie, and it was localized in the nucleus and cytoplasm. *BnbZIP2* transcripts were higher in male and female flowers, and its expression was induced by drought and ABA treatment. Plants with overexpression of *BnbZIP2* showed high sensitivity to Cd stress, especially at the seed germination stage. Further analysis of this TF can provide significant results for Cd stress tolerance in ramie. Hence, this paper recommends further research and analysis of this TF in future studies [156].

MYB transcription factors (TFs) play a key role in plants and are accountable for numerous biological procedures such as growth, metabolism, and response to metals stress. The MYB TFs have been widely investigated in numerous plant species. Nevertheless, complete data on MYB TFs in ramie remains a mystery [30]. In another TFs analysis, 105 BnGR2R3-MYB genes were recognized from the ramie genome and classified into 35 subfamilies based on phylogeny discrepancy and sequences resemblance. Remarkably, the expression of *BnGMYB10/12/41* in stems, roots, and leaves of ramie roots enhanced more than 10-fold after Cd stress, and they may cooperate with critical genes controlling the biosynthesis of flavonoids. Thus, protein communication network investigation recognized a possible connection between responsiveness to Cd stress and flavonoid biosynthesis. In theory, the studies enhanced the understanding of the valuable roles of MYBs and provided a basis for studying the genes in ramie crops. Hence, more studies are required to investigate the role of potential TF, *BnGMYB10/12/41*, regarding its interaction with other genes, especially genes regulating flavonoid biosynthesis to regulate Cd tolerance in ramie. In this way, researchers can generate more genetic material for further studies regarding the MYB TFs family and its role in Cd tolerance [30]. MYB TFs have been studied for their crucial role in Cd tolerance in many crops, especially ramie and *Arabidopsis*. An MYB gene called *BnMYB2* was cloned from ramie, enhancing Cd tolerance in *Arabidopsis* [37]. Thus, these TFs may regulate Cd stress and can be used to develop tolerant ramie varieties or other industrial crops for higher yields. Further analysis indicated that this TF has a close phylogenetic association with other 1R-MYB TFs which play a role against diverse environmental stresses. Hence, further characterization of this TF is missing and demands further study to accelerate Cd tolerance breeding [37].

The *bnMYB3* gene cloned from the ramie genome by amplifying cDNA ends (RACE) showed significant tolerance to Cd stress. The results of real-time quantitative PCR examination demonstrated that the *BnMYB3* gene was a constituent. Still, its expression levels in leaves and stems were more significant than in roots. Furthermore, *BnMYB3* expression was seemingly stimulated by Cd, and the relative expression level of the *BnMYB3* gene intensely improved with the continuation of Cd stress time and the increased concentration of $Cd^{2+}$. Further characterization of this gene can lead to identifying different roles against

Cd stress. Hence, it is recommended to conduct more studies to identify the pathways regulated by the *BnMYB3* gene in ramie [157]. Recently, another MYB TF, *BnMYB1*, was isolated from the ramie plant by rapidly amplifying cDNA ends based on the expression profiling of Cd-tolerant genes. Additionally, the *BnMYB1* gene was up-regulated after Cd stress, and the expression pattern of *BnMYB1* altered along with the Cd concentration and stress time. Collectively, our data proposed that *BnMYB1* (Table 3) is a cadmium-responsive factor and may show possible roles in plant's adjustment to Cd stress. This is a possible candidate for Cd-tolerant breeding, and further studies can authenticate its role against Cd stress and other heavy metals. Its expression level can be enhanced in various plant parts, including the leaves, roots, and stems [158].

**Table 3.** Potential TFs genes for enhancing Cd tolerance in ramie.

| TFs Families/Genes | Gene | Function | Reference |
|---|---|---|---|
| MYB | *BnGMYB10/12/41* | Interact with genes regulating the biosynthesis of flavonoids and enhanced Cd tolerance | [30] |
| MYB | *BnMYB2* | Enhanced Cd tolerance in Arabidopsis | [37] |
| MYB | *BnMYB3* | Higher expression levels with an increase in stress time and intensity | [157] |
| MYB | *BnMYB1* | Up-regulated and higher expression in leaves | [158] |
| bZIP | *BnbZIP3* | Enhanced Cd tolerance | [38] |
| bZIP | *BnbZIP2* | Enhanced sensitivity to Cd stress | [156] |

Pathway enrichment analysis exposed that the DEGs enhanced the cutin and wax biosynthesis pathway in ramie. Detecting these Cd-tolerant genes and pathways will be supportive in developing Cd-tolerant ramie cultivars. Further genetic analysis is critical to identify the pathways controlled by these identified genes. Cutin and way biosynthesis genes should be used in molecular breeding programs in future studies [59]. NAC is one of the significant TFs families, but the number of identified and cloned genes is minimal. In an experiment, 32 full-length BnNAC genes and 47 known NAC proteins were identified in other species. Alongside these, 32 proteins were used for phylogenetic analysis. The results showed that seventy-nine NAC family proteins could be classified into eight groups. Additionally, the expression of BnNAC genes was controlled by Cd stress, respectively. The documentation of these 32 full-length BnNAC genes and the description of their expression outline offer a prospect for the future explanation of their roles in ramie growth and development. These genes could be used for Cd tolerance in ramie to perform better in contaminated soils. The possible gaps in this study suggested that further studies are required to clarify the functions of these genes in ramie growth and development. Furthermore, NAC is not a highly studied TFs family, and future researchers should complete the functional characterization of this family regarding its role in Cd tolerance in ramie [159]. After a comprehensive analysis of available data on TFs for Cd tolerance in ramie, it can be concluded that this information is insufficient to adopt a reliable breeding tool to counter the threat of Cd stress on ramie growth and development. Different TFs families such as DREB and WRKY have not been profoundly investigated in ramie regarding Cd tolerance. The identification of TFs genes, their cloning, and transformation will increase Cd tolerance and lead to the development of tolerant ramie cultivars. It is important to expose the ramie to different levels of Cd stress at different growth stages and identify the potential TFs regulating Cd tolerance. This comprehensive review compiled all available data on the role of TFs in Cd tolerance in ramie, highlighted research gaps, and recommended further studies.

## 8. CRISPR/Cas9-Mediated Gene Editing for Cd Tolerance

The CRISPR/Cas9-mediated gene editing system is discovered in the immune system of bacteria and archaea [160]. It comprises short RNA sequences and endonucleases that

cleave the genome of foreign invaders at targeted points [161]. CRISPR/Cas9 comprises sgRNA and Cas proteins, which act as biological scissors [162]. CRISPR/Cas9 systems have been divided into two major groups: class 1, which has an extensive multiple Cas protein network for alignment and cleavage. The second group is class 2, which employs a single protein for editing [163]. The class 2 CRISPR program has been widely used in biological research because of the advantages offered by a single protein in the group. The class 2 CRISPR system has been further divided into three types, II, V, and VI, in which the nucleases are Cas9, 12a, and 13 [164]. Most Cas9 and Cas12 variants work in RNA-dependent DNA slicing that can encourage double-stranded breaks (DSBs), although Cas13 types have an RNA-dependent RNA cleavage action [165,166]. Newly emerged gene editing systems, such as base editing (BE) and prime editing (PE), can be used to edit large DNA parts [167]. These systems have further expanded the scope and boundary of the CRISPR/Cas9 system, but their use is not limited. It is strongly believed that use of BE and PE systems will increase with time [167,168].

The CRISPR/Ca9 technique has been practiced to enhance heavy metal tolerance in crops [169]. CRISPR/Ca9 has emerged as a promising tool for the targeted gene editing of desirable traits [170]. The CRISPR/Cas9 system is capable of developing environmentally resilient crop cultivars [171]. CRISPR/CCas9 has been successfully used in crops, animals, and bacteria for precise gene editing [172–174]. To avoid the off-target effects during gene editing, several web tools have been developed to design single-guided RNA (sgRNA) [175,176]. A web tool, CRISPR-P, has been established for sgRNA in 20 plant species [177]. Several studies have employed a detailed protocol for a targeted mutation in crops using CRISPR/Cas9 [178,179], and a toolkit and some vectors have been developed for CRISPR/Cas9 [180,181]. These studies have offered the use of CRISPR-Cas9 for many purposes, including genetic screening, editing, and transcriptional modulation to unfold the molecular mechanism of abiotic stress tolerance and to breed stress-tolerant crops [182].

Although CRISPR/Cas9 has generated several successful stories about its use in targeted gene editing in many crops [183,184], there is no evidence of CRISPR/Cas9 for Cd tolerance in ramie. CRISPR/Cas9 has been employed in many crops for the development of Cd tolerance. A successful example was presented by Liu et al. [185], in which they edited the Sl1 gene in tomatoes using CRISPR/Cas9, which enhanced Cd tolerance by increasing the efficiency of antioxidant enzymes and increased photosynthesis. Gene expression analysis showed that *Sl1* regulated the transcript levels of heavy metals transporter genes to prevent Cd accumulation [185]. Many heavy metal transporters have been knocked out using CRISPR/Cas9 to enhance abiotic tolerance in many crops. A heavy metal transporter, *OsABCG36*, is responsible for rice's Cd tolerance. The CRISPR/Cas9-mediated knockout of *OsABCG36* increased Cd tolerance in rice plants by transporting Cd from root cells in rice [186]. There has been no report of CRISPR/Cas9-mediated gene editing for Cd tolerance in ramie. Many potential genetic resources offer an excellent opportunity for successfully using CRISPR/Cas9 to use targeted gene editing in ramie for Cd tolerance. The CRISPR/Cas9-mediated targeted gene editing of metal transport and other genes from wild types and landraces can be helpful to develop Cd-tolerant ramie cultivars to sustain growth and development in Cd-polluted soils. It is important to screen the ramie germplasms for the identification of stress responsive genes, the extraction of the genome, the construction of the gene library, and the use of potential molecular tools for successful genetic modification in the desired gene for desired traits. CRISPR/Cas9 can also be used to enhance the genetic diversity in ramie through the creation of mutations by artificial means in available germplasm.

## 9. Conclusions and Future Perspectives

Agriculture plays a crucial role in providing the dietary needs of all human beings. Plant breeding aims to increase the yield of crops to maintain balance in food security. However, different types of biotic and abiotic stresses have significantly affected the growth and yield of many crops, leading to increased hunger in many parts of the world. Cd is a

non-essential heavy metal that has no biological role in plants, damaging ramie growth and production even at low concentrations. Cd toxicity effects can be seen at the whole-plant, cellular, and molecular levels, reducing photosynthesis, protein synthesis, introducing retardation of growth, nutritional disorder, and an imbalance in the plant water relation. Ramie is China's most significant fiber crop, and has been badly affected by Cd toxicity and requires special attention from researchers. Plants use several defense mechanisms to counter the toxic effect of Cd, such as the activation of the antioxidant defense system, as well as the synthesis of genes, proteins, and other stress-signaling pathways. The Cd tolerance mechanism is complex and governed by many genes and their regulatory networks. Plant breeders have been trying to develop Cd tolerance in ramie by conventional breeding methods; however, the large-scale use of these methods is limited. QTL mapping using diverse populations should be enhanced to identify potent regions involved in Cd tolerance and their transformation using QTL pyramiding. Previous genetic engineering studies have led to the development of many Cd-tolerant genotypes in many crops; however, the use of this tool in ramie is negligible.

TFs technique has been widely used in ramie to identify the genes overexpressed under Cd stress, and they have been classified based on their families. We strongly suggest using the identified TFs and QTL in molecular breeding programs. CRISPR/Cas9-mediated gene editing is a novel tool that broke biological barriers and has successfully edited the many abiotic tolerance genes in the crops. Unfortunately, the application of CRISPR/Cas9 based gene editing for Cd tolerance in ramie has not been reported, which offers an excellent opportunity to apply this tool for developing Cd-tolerant ramie cultivars. Agronomic techniques such as using different growth-promoting hormones, such as GA, CK, ET, JA, and ABA, are mandatory to counterbalance the toxic effect of Cd stress in ramie. The enhancement of the antioxidant defense mechanism is critical to naturalize ROS and MDA production under Cd stress. Ramie breeders should conserve the genetic diversity of ramie to increase the efficiency of Cd-tolerant breeding in future studies.

We strongly recommend the following steps to develop Cd-tolerant ramie cultivars

- The screening of ramie germplasm will lead to the identification of several tolerant genotypes, which can serve as a source of Cd-tolerant genes for developing Cd-tolerant ramie.
- Investigation of physiological- and biochemical-based Cd tolerance mechanisms.
- Selection of wild relatives of ramie and their screening for Cd-tolerant genes.
- Exposure of ramie cultivars to Cd stress levels and increase the expression of TFs families and the selection of potential targets for CRISPR/Cas9-mediated gene editing to develop Cd-tolerant mutants.
- Conservation of the genetic diversity of ramie for future breeding programs.
- Increase the Cd accumulation in ramie to increase its phytoremediation capability.

**Author Contributions:** A.R. prepared the manuscript, H.J., Y.M., P.H., H.X. and X.L. helped in review and editing. M.U.H., S.F.A.G., B.A., R.O.E., M.U.H. and S.A. reviewed the manuscript. Y.J. supervised the study and provided funding. All authors have read and agreed to the published version of the manuscript.

**Funding:** This study was financially supported by the National Natural Science Foundation of China (32071940), China National Key R&D Program (2019YFD1002205-3 and 2017FY100604-02), and Foundation for the Construction of Innovative Hunan (2020NK2028).

**Data Availability Statement:** Not applicable.

**Acknowledgments:** The authors thank the Deanship of Scientific Research, King Khalid University for supporting this work through the research groups program under grant number R.G.P. 2/75/44.

**Conflicts of Interest:** The authors declare no conflict of interest.

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
