# Peer review of "A Systematic Review on the Improvement of Cd Stress Tolerance in Ramie Crop, Limitations and Future Prospective"

_agronomy, doi:10.3390/agronomy13071793_

Round 1

Reviewer 1 Report

The paper is well-structured, interesting, and readable. However, the scientific contribution of the paper seems limited in some parts, and then it should be improved. The scientific contribution of the paper is unclear: the theoretical and practical implications are rather partial. Also, the technical challenges are quite unclear. Not enough technical details were provided.

In my opinion, the introduction should be further improved to better specify the work's motivation and the methodology's characteristics. As it is, it is not clear how it works. Additionally, the author should state clearly in which aspect this work extends the state of the art, i.e., highlight the novelty…

About the literature review, each paper should clearly specify what is the proposed methodology, novelty and results with experimentation. At the end of related works, highlight better in some lines what overall technical gaps are observed in existing works that led to the design of the proposed approach. To better delineate the context and the different possible solutions, you can consider the following papers as references: http://www.jsoftware.us/vol1/jsw0102-02.pdf and https://www.hindawi.com/journals/jhe/2022/3861161/.

Some clarifications are needed in the experimental section. Further information is required. The analysis and interpretation of the test findings need to be improved.  Finally, the results section should be correlated with a discussion phase to validate the proposed thesis

Author Response

Reviewer 1

Comments and Suggestions for Authors

The paper is well-structured, interesting, and readable. However, the scientific contribution of the paper seems limited in some parts, and then it should be improved. The scientific contribution of the paper is unclear: the theoretical and practical implications are rather partial. Also, the technical challenges are quite unclear. Not enough technical details were provided.

R: Dear reviewer, thanks for your comments. We have improved the scientific contribution of paper in some sections which were not clear. Please see conclusion section of each section and especially, conclusion of paper. Ramie is not a highly studied crop. This review provides an updated picture of recent advancements on Cd tolerance in ramie and will serve as a potential source of information to conduct further studies.

In my opinion, the introduction should be further improved to better specify the work's motivation and the methodology's characteristics. As it is, it is not clear how it works. Additionally, the author should state clearly in which aspect this work extends the state of the art, i.e., highlight the novelty…

R: Dear reviewer, thanks for comments. We have improved the introduction section and explained the novelty in a better way.

About the literature review, each paper should clearly specify what is the proposed methodology, novelty, and results with experimentation. At the end of related works, highlight better in some lines what overall technical gaps are observed in existing works that led to the design of the proposed approach. To better delineate the context and the different possible solutions, you can consider the following papers as references: http://www.jsoftware.us/vol1/jsw0102-02.pdf and https://www.hindawi.com/journals/jhe/2022/3861161/.

R: We have tried our best to explain the related aspects of each paper and we also provided conclusion at end of each section and highlighted the research gap for future studies. You can study on QTL mapping and CRISPR/Cas9 for Cd tolerance are limited in ramie. We recommended to carry out these studies to address the Cd toxicity issues. In these sections we did not propose any study gap because there is already a huge gap to study these aspects in ramie. For example, in the section TFs analysis for Cd tolerance in ramie, we have written that ‘’.  Hence, more studies are required to investigate the role of potential TF, BnGMYB10/12/41 regarding its interaction with other genes and especially genes regulating flavonoid biosynthesis to regulate the Cd tolerance in ramie. In this way, researchers can generate more genetic material for further research studies about MYB TFs family and its role in Cd tolerance’’. These statements indicated the research gaps in existing studies and highlighted that further studies are required.

Some clarifications are needed in the experimental section. Further information is required. The analysis and interpretation of the test findings need to be improved.  Finally, the results section should be correlated with a discussion phase to validate the proposed thesis.

R: This is reviewing paper and does not involve results and discussion section. We already discussed the papers findings in each section. We have added further data in each section.

Reviewer 2 Report

The present paper  is certainly very interesting, dealing with a relevant problem from an agronomic, chemical and biological point of view. However there are some statements that should be better clarified.

A) What are the causes indicated here: "however, due to the continuous rise in Cd toxicity, the use of potent molecular tools is critical in modern day breeding programs".

In particular where and in which specific situations there is an increase of Cd(II) ions in the agricultural ecosystem.

B) The article analyzes the role of plant hormones in modulating the process of tolerance to Cd(II) ions, but does not report the role of giberellin, one of the most efficient hormones capable of modulating the effects of Cd(II) ions .

"Each hormone's mechanism of action may differ for each crop. It has been studied that multiple hormones may regulate a single process, and simultaneously, different processes are controlled by a single hormone. Plant hormones like auxin, abscisic acid (ABA), cytokinin (CK), ethylene (ET), and jasmonic acid (JA) are important plant hormones that are vital for plant growth and development".

In the paper there are some syntax and grammatical errors, as well as some misspellings.

Author Response

Reviewer 2

Comments and Suggestions for Authors

The present paper is certainly very interesting, dealing with a relevant problem from an agronomic, chemical, and biological point of view. However, there are some statements that should be better clarified.

  1. What are the causes indicated here: "however, due to the continuous rise in Cd toxicity, the use of potent molecular tools is critical in modern day breeding programs".

R: Thanks for your comments. We have added the statement that ‘’conventional breeding tools have limited applications in Cd-tolerant breeding because of the complex genetic mechanism of Cd tolerance. This is the reason we use molecular tools to unpin the genetic mechanism behind Cd tolerance in ramie.  Please see the introduction section.

Where and in which specific situations there is an increase of Cd (II) ions in the agricultural ecosystem.

R:  We have added this information in introduction section. Cd2+ is divalent form of Cd which exists in soils at concentrations typically ranging 0.1–1.0 mg kg−1. Cd ions are increased in soil due to anthropogenic activities like, mining, atmospheric deposition of combustion emissions, mining, sewage sludge, and the use of Cd-containing fertilizers.

  1. B) The article analyses the role of plant hormones in modulating the process of tolerance to Cd (II) ions, but does not report the role of gibberellin, one of the most efficient hormones capable of modulating the effects of Cd (II) ions. “Each hormone's mechanism of action may differ for each crop. It has been studied that multiple hormones may regulate a single process, and simultaneously, different processes are controlled by a single hormone. Plant hormones like auxin, abscisic acid (ABA), cytokinin (CK), ethylene (ET), and jasmonic acid (JA) are important plant hormones that are vital for plant growth and development".

R: Dear reviewer, thanks for your comments. We only find one study about the role of GA in Cd tolerance in ramie. In earlier study, ‘The results indicated that, GA-1 enhanced the Cd content of the aboveground ramie to 3 times more than that of the control and decreased the Cd content of the underground ramie by 54.76%’. Basically, we want to develop the ramie cultivar with super-accumulating nature for phyto-remediation of Cd-polluted nature. We have already added the role of IAA, ABA, and JA in Cd tolerance in ramie. Meanwhile, we have added some other techniques dealing with the improvement of Cd tolerance in ramie.

Reviewer 3 Report

Dear authors,

The manuscript ID agronomy-2439624 is a very interesting review and it is well organized.

I suggest to mention the word "review" from the title in order to be easier to have an idea about the content.

The list of references cannot be exhaustive but can be extended, as well as the length of the manuscript, as it is a review article.

Good luck!

Author Response

Reviewer 3

Comments and Suggestions for Authors

Dear authors,

The manuscript ID agronomy-2439624 is a very interesting review and it is well organized.

R: Dear reviewer, thanks for your comment.

I suggest to mention the word "review" from the title in order to be easier to have an idea about the content.

R: Dear reviewer, thanks for comments. The word ‘review’ is already added in ‘title’.

The list of references cannot be exhaustive but can be extended, as well as the length of the manuscript, as it is a review article.

R: Thanks for comment. We have added more data in paper and also increased the length of references.